



# Subglacial valleys preserved in the highlands of south and east Greenland record restricted ice extent during past warmer climates

Guy J. G. Paxman[1], Stewart S. R. Jamieson[1], Aisling M. Dolan[2], Michael J. Bentley[1]

[1]Department of Geography, Durham University, Durham, DH1 3LE, United Kingdom

[2]School of Earth and Environment, University of Leeds, Leeds, LS2 9JT, United Kingdom

*Correspondence to*: Guy J. G. Paxman (guy.j.paxman@durham.ac.uk)

**Abstract.** The Greenland Ice Sheet is a key contributor to contemporary global sea level rise, but its long-term history and response to episodes of warming in Earth's geological past remain uncertain. The terrain covered by the ice sheet comprises ~79% of Greenland and ~1.1% of the Earth's land surface and contains geomorphological records that may provide valuable

insights into past ice-sheet behaviour. Here we use ice surface morphology and radio-echo sounding data to identify ice-covered valleys within the highlands of southern and eastern Greenland and use numerical ice-sheet modelling to constrain the climatological and glaciological conditions responsible for valley incision. Our mapping reveals intricate subglacial valley networks with morphologies that are indicative of substantial glacial modification of an inherited fluvial landscape, yet many of these valleys are presently situated beneath cold-based, slow-moving (i.e., non-erosive) ice. We use the morphology of the

valleys and our simple ice-sheet model experiments to infer that incision likely occurred beneath erosive mountain valley glaciers during one or more phases of Greenland's glacial history when ice was restricted to the southern and eastern highlands, and Greenland's contribution to barystatic sea level was up to +7 metres relative to today. We infer that this valley incision primarily occurred prior to the growth of a continental-scale ice sheet, most likely during the late Miocene (ca. 7–5 Ma) and/or late Pliocene (ca. 3.6–2.6 Ma). Our findings therefore provide new data-based constraints on early Greenland Ice Sheet extent

and dynamics that can serve as valuable boundary conditions in models of regional and global palaeoclimate during past warm periods that are important analogues for the 21st Century and beyond.

## 1 Introduction

Loss of ice from the Greenland Ice Sheet (GrIS) is one of the largest contributors to contemporary sea level rise (Cazenave et al., 2018). Between 2005 and 2018, the GrIS contributed an average of 0.76 mm yr$^{-1}$ to global mean sea level rise (Cazenave

et al., 2018) due to mass loss driven by increased surface melting and enhanced discharge from outlet glaciers (The Imbie Team, 2020). Climate-ice sheet model ensemble experiments indicate that rates of ice mass loss in Greenland are likely to accelerate over the course of the current century, with the GrIS contributing several centimetres of additional global mean sea level rise by 2100 (Aschwanden & Brinkerhoff, 2022; Edwards et al., 2021; Goelzer et al., 2020). However, predicting future GrIS behaviour remains challenging due to incomplete understanding of ice dynamics, including the complex interactions



between the ice sheet and the atmosphere, solid Earth, and ocean (Alley et al., 2019; Goelzer et al., 2013; Straneo & Heimbach, 2013). A key means of improving our understanding of current and future GrIS behaviour is to examine records of past behaviour. However, significant uncertainties persist regarding the long-term history of the GrIS, particularly its extent and behaviour during past warm periods of the Quaternary and Neogene (Fig. 1a).

Given that ice covers approximately 79% of the land surface of Greenland (Fig. 1b), many of the records used to infer long-
term past ice-sheet behaviour have been acquired via coring or seismic imaging of offshore sedimentary material. For example, the presence of ice-rafted debris in North Atlantic and Arctic shelf and deep ocean sediments suggests that near-coastal ice existed in eastern Greenland from the late Miocene (ca. 11–7 Ma) (Helland & Holmes, 1997; Larsen et al., 1994; St. John & Krissek, 2002) and potentially as early as the late Eocene (ca. 38 Ma) in ephemeral form (Eldrett et al., 2007; Tripati & Darby, 2018) (Fig. 1a).

Records of ice-rafted debris in the North Atlantic (Bailey et al., 2013; Blake-Mizen et al., 2019; Flesche Kleiven et al., 2002; Jansen et al., 2000) and global ice volume signals extracted from marine oxygen isotope records (Mudelsee & Raymo, 2005) have been used to infer that the gradual transition towards large-scale glaciation in Greenland — referred to as the 'onset of Northern Hemisphere Glaciation' (McClymont et al., 2023) — occurred during the late Pliocene to earliest Pleistocene (ca. 3.6–2.4 Ma; Fig. 1a). Glaciation is thought to have been primarily controlled by a drop in atmospheric $CO_2$ levels to below a
key threshold (DeConto et al., 2008; Lunt et al., 2008), although other global factors such as uplift of the Himalayas and Rocky Mountains (Ruddiman & Kutzbach, 1989), closure of the Panama seaway (Haug & Tiedemann, 1998), and/or shifts in atmosphere and ocean circulation (Wara et al., 2005) may have also been partially responsible (Lunt et al., 2008). The initial phase of ice growth was likely interrupted by the mid-Piacenzian warm period (mPWP; ca. 3.26–3.02 Ma) (Haywood, Dowsett, & Dolan, 2016); the global sea level highstand of up to +25 m during this interval (Dumitru et al., 2019; Dutton et al., 2015;
Grant et al., 2019) implies that the GrIS was likely significantly smaller than at the present-day.



**Figure 1: Greenlandic glacial history.** (a) Cenozoic global benthic oxygen isotope curve (5-point running mean) (Westerhold et al., 2020) (Eoc. = Eocene; Olig. = Oligocene). Note the logarithmic age scale. Red bars mark notable Plio-Pleistocene warm intervals (MIS = Marine Isotope Stage). (b) Present-day Greenland Ice Sheet. Hillshade image of the ice surface (Howat et al., 2014, 2022) is shaded in grey; land surface topography and seafloor bathymetry are displayed according to the colour scale (relative to mean sea level) (Morlighem et al., 2017). Black lines demarcate major ice-sheet drainage basins (Zwally et al., 2012); blue triangles mark deep ice core sites. Red dashed line marks the GrIS extent used as a PRISM4 (Pliocene Research Interpretation and Synoptic Mapping) palaeogeography boundary condition representing the mPWP interval in the Pliocene Model Intercomparison Project Phase 2 (Dowsett et al., 2016; Haywood, Dowsett, Dolan, et al., 2016).

The intensification of large-scale glaciation on Greenland is believed to have commenced following the mPWP, culminating in a continental-scale ice sheet by the latest Pliocene / earliest Pleistocene (ca. 2.7–2.4 Ma) (Christ et al., 2020; Flesche Kleiven




et al., 2002; Lisiecki & Raymo, 2005; McClymont et al., 2023). Interglacial marine sediments and terrestrial material exposed along the northern Greenland coast indicate that periods characterised by boreal forest-tundra conditions and less extensive ice
cover persisted (at least intermittently) into the early Pleistocene (Bennike et al., 2010; Feyling-Hanssen et al., 1983; Funder et al., 2001). Dynamic behaviour of the GrIS in the early Pleistocene is also evidenced by offshore seismic reflection data, which reveal major cross-shelf glacial troughs and progradational trough-mouth fans and sedimentary wedges that are indicative of multiple phases of ice sheet shelf-edge advance and retreat (Knutz et al., 2019; Nielsen & Kuijpers, 2013).

Cosmogenic radionuclide (e.g., $^{10}$Be and $^{26}$Al) concentrations in sediment and bedrock obtained via sub-ice drilling at the
GISP2 and Camp Century sites (Fig. 1b) indicate that these sites have not been continuously covered by ice for more than approx. 1.1 Myr, suggesting that at least one episode of major ice loss occurred in northwest and central Greenland at or since 1.1 Ma (Christ et al., 2021; Schaefer et al., 2016). However, cosmogenic radionuclides in North Atlantic sediments indicate that ice cover persisted at least in eastern Greenland throughout this time (Bierman et al., 2016). Candidate time intervals for partial late Pleistocene deglaciation include particularly warm or long-lasting interglacial periods, such as Marine Isotope
Stage (MIS) 31 (ca. 1.09–1.06 Ma) (Melles et al., 2012) or MIS 11 (ca. 430–400 ka) (Christ et al., 2023; Reyes et al., 2014; Robinson et al., 2017) (Fig. 1a). Most recently, the GrIS underwent a period of more muted retreat during the Last Interglacial Period (the Eemian; MIS 5e; ca. 125 ka) (Plach et al., 2018), before expanding onto the continental shelf at the Last Glacial Maximum (LGM; ca. 21 ka) (Lecavalier et al., 2014) and subsequently retreating to its modern configuration through the Holocene (Lesnek et al., 2020).

Building an improved understanding of GrIS behaviour during its early development and through subsequent warmer climate intervals will provide valuable analogues for predicting its response to current and projected future atmosphere and ocean warming. Model intercomparison projects such as PLISMIP (Pliocene Ice Sheet Model Intercomparison Project) demonstrate that there is significant model spread in the predicted GrIS configuration during past warm periods such as the mPWP, which largely relates to uncertainties in the climate forcing (Dolan et al., 2015; Koenig et al., 2015). This highlights the need for more
data-based constraints of past ice extent and behaviour. Unfortunately, evidence for long-term GrIS behaviour is limited in quantity and irregularly distributed, reflecting the logistical challenges associated with acquiring geological data, the masking of 79% of the land surface by the GrIS, and the fact that as the ice sheet waxes and wanes it erases the onshore debris that would otherwise provide a record of previous glacial cycles.

In areas elsewhere on Earth formerly covered by large ice sheets, investigations of large-scale glacial geomorphology have
provided important insights into past ice-sheet extent and behaviour, given that the landscape typically records the long-term average glacial conditions to which it has been subjected (Porter, 1989). This paradigm has recently begun to be applied to the bed topography of Greenland and Antarctica, which has been measured/inferred using the large quantities of radio-echo sounding (RES) survey data and ice surface morphology imagery acquired in recent years (Haran et al., 2018; MacGregor et al., 2021). Previous studies have shown that the subglacial landscape records processes including glacial erosion, fluvial



incision in the absence of ice, tectonic deformation, and isostatic adjustment (Cooper et al., 2016; Livingstone et al., 2017; Paxman et al., 2021; Pedersen et al., 2019). The topography of the 79% of Greenland that is covered by the GrIS can therefore provide a valuable, and hitherto relatively little exploited, opportunity for understanding past ice-sheet behaviour, particularly during intervals associated with smaller-than-present ice configurations. Significantly, ice sheets can preserve large areas of topography beneath cold-based, non-erosive ice, meaning that geomorphological records pertaining to the early development

of the ice sheet can survive for millions of years (Bierman et al., 2014; Rose et al., 2013).

The aim of this study is to use ice surface morphology and RES survey data to map and interpret the subglacial landscape of the highlands of south and east Greenland (Fig. 1b). We focus on this area because these highlands are believed to have been the site of initial GrIS inception (Solgaard et al., 2013) and to have retained ice during subsequent warm periods (Bierman et al., 2016; Schaefer et al., 2016), and are therefore the area most likely to contain geomorphological records of ice behaviour

during intervals when the GrIS was significantly more restricted in extent. We combine our geomorphological mapping with numerical ice-sheet modelling to understand the processes responsible for the evolution of the subglacial landscape of these highlands and the implications for long-term ice behaviour.

## 2 Methods

### 2.1 Subglacial valley mapping

We used a combination of satellite imagery and airborne RES survey data to map subglacial valleys present in the eastern and southern highlands of Greenland (Fig. 1b). The distribution and morphology of valleys and their networks (i.e., how they are spatially organised and connected) are particularly useful in establishing the patterns and scales of past erosion, and in turn the dynamics of past ice sheets (Livingstone et al., 2017; Rose et al., 2013).

### 2.1.1 Ice surface morphology

We followed the approach of previous studies (Jamieson et al., 2023; Lea et al., 2023; Ross et al., 2014) that have used ice surface morphology, as recorded by satellite imagery, to infer the locations of subglacial valleys and ridges. For this task, we used the 100 m-resolution MODIS (Moderate Resolution Imaging Spectroradiometer) Mosaic of Greenland (MoG) surface morphology image map version 2 (Haran et al., 2018). MODIS MoG imagery records the intensity of the reflection of a satellite-emitted radar signal, which depends on the ice surface slope / curvature. Depending on the local glaciological

conditions, subtle changes in surface slope can be predictably correlated with undulations in bed topography (Le Brocq et al., 2008; Rémy & Minster, 1997); typically, subglacial valleys are positioned below darker areas (lower reflection intensity) and subglacial ridges below lighter areas (higher reflection intensity) in the MoG image (Fig. 2). Surface morphology mapping is often ineffective close to ice divides, where ice is thickest and horizontal velocities are near-zero, meaning surface expressions of basal features are heavily attenuated. Conversely, isolating the signal from the basal topography is challenging in areas



characterised by high basal sliding rates and active supra- and en-glacial dynamics (such as ice streams), which also influence ice surface morphology (Cooper et al., 2019a). Surface imagery is therefore best suited to inferring qualitative variations in bed topography in areas of relatively slow (but non-zero) ice surface velocity, minimal basal sliding, high bed relief, and thin ice. The southern and eastern highlands of Greenland fulfil all of these criteria (Joughin et al., 2018; MacGregor et al., 2022; Morlighem et al., 2017) and are therefore an ideal target for this analysis.

### 130 2.1.2 Radio-echo sounding

The amplitude transfer of bed undulations to the ice surface is complex and dependent on basal sliding velocity, ice rheology, and ice thickness (Gudmundsson, 2003; Ng et al., 2018). Therefore, while useful for inferring the qualitative 'form' of a subglacial landscape, ice surface morphology alone cannot be used to establish the absolute elevation of the bed. RES surveys directly measure the ice thickness (and thus bed elevation) along flight lines; these data can be used to ground-truth the

locations of subglacial valleys and quantify their cross-profile morphology. For this task, we used a recently assembled database of the locations of subglacial valleys visible in Operation IceBridge RES data (Fig. S1a) across the GrIS (Paxman, 2023). This dataset also contains quantitative metrics of valley cross-profile morphology, including depth, width, V-shapedness, and curvature, as well as classifications of valleys as either 'glacial' or 'fluvial' based on their morphological similarity to glacial or fluvial valleys observed elsewhere in the Northern Hemisphere. This valley classification was

accomplished using the random forest supervised machine learning algorithm (Breiman, 2001), which is a statistical method associated with uncertainty that is encapsulated in the classification 'score'. For the purposes of this analysis, we only examined valleys with a score of at least 0.75, indicating that the classification is associated with relatively high confidence or probability (for more information the reader is referred to Paxman, 2023).

MODIS ice surface imagery and RES bed elevation data are therefore complementary, providing (i) an indication of the morphology of valley cross-profiles where they are intersected by flight lines and (ii) a view of valley planform geometry (i.e., orientation and connectivity) between flight lines (Fig. 2). Using the MODIS MoG image and the RES-derived subglacial valley database (Paxman, 2023), we digitised valley planform patterns within the highlands of southern and eastern Greenland visible in ice surface imagery, ground-truthed their locations using RES data, and extracted their cross-profile morphometrics.





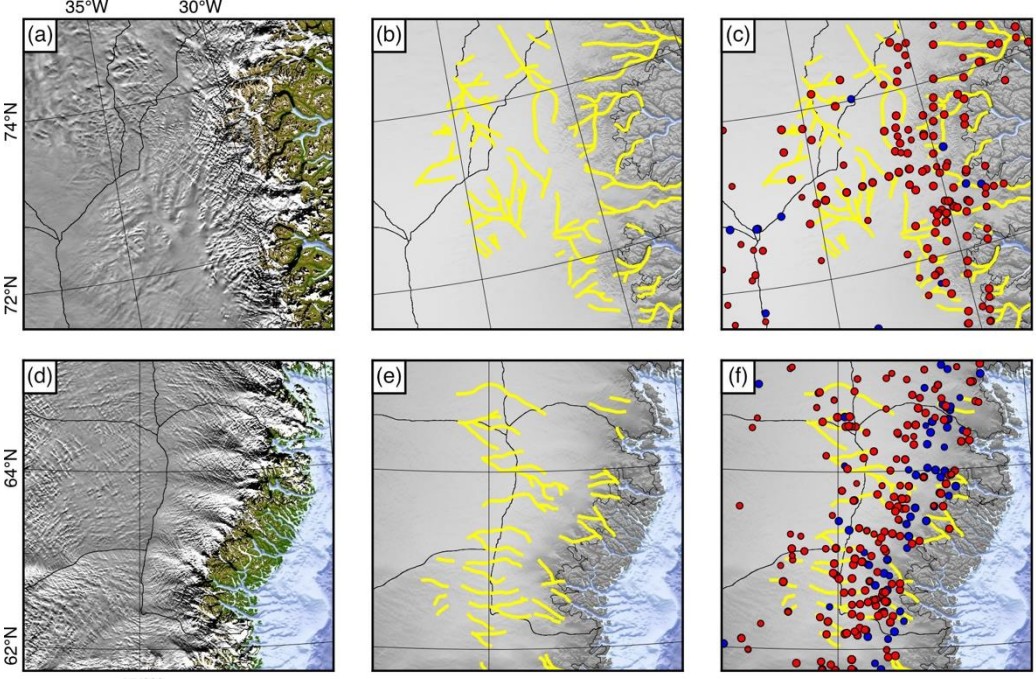

**Figure 2. Illustration of subglacial valley mapping from ice surface morphology imagery and RES survey data**. (a, d) MODIS MoG ice surface morphology image (Haran et al., 2018). (b, e) Locations of interpreted subglacial valleys (yellow) overlain on shaded ice surface elevation (Howat et al., 2022) over the same areas as shown in the panels on the left. (c, f) Valleys identified in RES survey data (Paxman, 2023) overlain on valley planforms mapped from MODIS MoG morphology imagery (yellow). Red circles = 'glacial' classification; blue = 'fluvial' classification. Panels a–c cover the area shown in Fig. 4c; panels d–f cover the area shown in Fig. 4e.

## 2.2 Ice-sheet modelling

The valleys in the southern and eastern highlands may have been formed, exploited, and/or modified during past mountain-scale glaciation in Greenland (Patton et al., 2016). They may therefore provide important constraints on the extent and behaviour of past ice sheets. We performed simple numerical ice-sheet modelling experiments to test whether it was possible to simulate ice masses with configurations consistent with the mapped valley extents and configurations, and if so to quantify the associated climatology. For this task we used the Parallel Ice Sheet Model (PISM), an open-source, finite-difference, shallow ice-sheet model (Winkelmann et al., 2011).

### 2.2.1 Model set-up

We used PISM in its 'hybrid' mode, which simulates internal deformation and basal sliding by employing the shallow-ice and shallow-shelf approximations (SIA and SSA), respectively. Our model set-up included constitutive 'laws' for temperature-dependent and water-content-dependent creep, which govern internal ice deformation (Aschwanden et al., 2012; Lliboutry & Duval, 1985; Paterson & Budd, 1982), and a pseudo-plastic sliding 'law' that governs basal sliding (Bueler & Brown, 2009; Winkelmann et al., 2011); we used a constant till friction angle of 30° based on analogy with the European Alps (Seguinot et



al., 2018). Unless otherwise stated, we used the PISM default values for the model parameters relating to ice rheology and
basal sliding (a full list of the parameters used in our simulations is provided in Table 1).

Since the focus of our modelling was on upland ice growth as opposed to (e.g.) grounding line dynamics at marine-terminating
glaciers, we assumed a simple calving model where ice is removed when it reaches the coastline; sea level was assumed to be
20 metres higher than today (Dutton et al., 2015). The model domain includes the entirety of Greenland with a horizontal grid
resolution of 10 km, which represents a compromise between computational efficiency and the ability to capture the larger
valley networks present in the subglacial topography (cf. Paxman, 2023). Selected experiments were also performed at a higher
resolution of 5 km, which reflects the average valley width in the eastern and southern highlands (Paxman, 2023), allowing us
to examine the influence of valley topography on the simulated patterns of ice flow in greater detail. Each simulation started
from an ice-free state, using the boundary conditions and climate forcing described below. In each case, the model was spun-
up for 10 kyr, after which ice extent and volume reached quasi-equilibrium in all experiments. Through the model run, the bed
topography was continually adjusted for isostatic deformation under the simulated ice load using an elastic lithosphere relaxing
asthenosphere model.

**Table 1: Parameters used in PISM simulations.**

| Name | Value | Unit |
|---|---|---|
| **Ice rheology** | | |
| Ice density | 910 | kg m$^{-3}$ |
| Acceleration due to gravity | 9.81 | m s$^{-2}$ |
| Glen exponent | 3 | - |
| Ice hardness coefficient (cold) | 3.61 x 10$^{-13}$ | Pa$^3$ s$^{-1}$ |
| Ice hardness coefficient (warm) | 1.73 x 10$^3$ | Pa$^3$ s$^{-1}$ |
| Flow law activation energy (cold) | 6.0 x 10$^4$ | J mol$^{-1}$ |
| Flow law activation energy (warm) | 13.9 x 10$^4$ | J mol$^{-1}$ |
| SIA enhancement factor | 3 | - |
| SSA enhancement factor | 1 | - |
| Flow law critical temperature | 263.15 | K |
| Flow law water fraction coefficient | 181.25 | - |
| Ideal gas constant | 8.31441 | J mol$^{-1}$ K$^{-1}$ |
| Clausius-Clapeyron constant | 7.9 x 10$^{-8}$ | K Pa$^{-1}$ |
| Ice specific heat capacity | 2009 | J kg$^{-1}$ K$^{-1}$ |
| Water specific heat capacity | 4170 | J kg$^{-1}$ K$^{-1}$ |
| Ice thermal conductivity | 2.1 | J m$^{-1}$ K$^{-1}$ s$^{-1}$ |



| Water latent heat of fusion | $3.34 \times 10^5$ | J kg$^{-1}$ K$^{-1}$ |
|---|---|---|
| **Basal sliding** | | |
| Pseudo-plastic sliding exponent | 0.25 | - |
| Pseudo-plastic threshold velocity | 100 | m yr$^{-1}$ |
| Till cohesion | 0 | Pa |
| Till reference void ratio | 0.69 | - |
| Till compressibility coefficient | 0.12 | - |
| Till effective fraction overburden | 0.02 | - |
| Till friction angle | 30 | ° |
| Maximum till water thickness | 2 | m |
| **Bedrock and lithosphere** | | |
| Bedrock density | 2700 | kg m$^{-3}$ |
| Bedrock specific heat capacity | 1000 | J kg$^{-1}$ K$^{-1}$ |
| Bedrock thermal conductivity | 3 | J m$^{-1}$ K$^{-1}$ s$^{-1}$ |
| Lithosphere flexural rigidity | $5 \times 10^{24}$ | N m |
| Mantle density | 3300 | kg m$^{-3}$ |
| Mantle viscosity | $1 \times 10^{21}$ | Pa s |
| **Climate** | | |
| Temperature of rain precipitation | 275.15 | K |
| Temperature of snow precipitation | 273.15 | K |
| PDD factor for ice | $8.791 \times 10^3$ | m K$^{-1}$ day$^{-1}$ |
| PDD factor for snow | $3.297 \times 10^3$ | m K$^{-1}$ day$^{-1}$ |
| Snow refreezing fraction | 0.6 | - |
| Ice refreezing fraction | 0 | - |
| Standard deviation of daily temperature variation in the PDD model | 5 | K |
| Air temperature lapse rate | -6.5 | K km$^{-1}$ |
| Precipitation exponential scaling factor | 0.0704 | K$^{-1}$ |
| Ice-free thickness limit | 10 | m |
| Sea level (relative to modern) | 20 | m |



### 2.2.2 Boundary conditions

The main subglacial boundary conditions required by the ice-sheet model are the geothermal heat flux and bed topography. We used a present-day geothermal heat flux model derived from magnetic anomaly data (Martos et al., 2018), and, in the absence of a robust reconstruction of palaeo-geothermal heat flux, assumed that the heat flux has remained constant since early ice growth. For the bed topography, we adjusted the modern BedMachine v.5 digital elevation model (Morlighem et al., 2017) for the isostatic response to (a) the complete removal of the modern GrIS (Paxman et al., 2022) and (b) erosional unloading

driven by Pleistocene fjord incision around the continental margins (Medvedev et al., 2013; Pedersen et al., 2019). For the erosional unloading correction, we used the approach described by Pedersen et al. (2019), whereby the fjords are filled to the level of the adjacent plateaux, and the isostatic response is computed using an elastic plate model with a laterally variable effective elastic thickness (Steffen et al., 2018). This correction was necessary given that we were attempting to simulate the growth of early ice sheets prior to fjord incision. However, while we adjusted regional elevations for the regional flexural

response to valley/fjord incision, we left the valleys themselves open (i.e., unfilled) in the reconstructed digital elevation model, enabling us to simulate ice flow through the valleys.

Because there is significant uncertainty surrounding both the past geothermal heat flux and the bed topography (and given that we are not modelling a specific past time interval), we used sensitivity testing to examine the influence of these boundary conditions on the ice-sheet model results. To do so, we ran sensitivity experiments using the heat flux and topography described

above and an alternative scenario for each boundary condition. The alternative heat flux map was derived by quantifying statistical relationships between global geological data and geothermal heat flux measurements to predict heat flux across Greenland from geophysical observations (Rezvanbehbahani et al., 2017). We are agnostic with regards to the relative merits of the two models, but importantly for our purposes they were determined using independent datasets and methodologies and contain contrasting estimates of heat flux for eastern and southern Greenland (30–50 mW m$^{-2}$ in Rezvanbehbahani et al. (2017)

versus 60–70 mW m$^{-2}$ in Martos et al. (2018)), allowing us to fully assess the sensitivity of our results to this parameter. The alternative topography was generated using the same isostatic correction for modern ice-sheet unloading, but the flexural response to valley incision was reduced by a factor of two (i.e., equivalent to conservatively filling the fjords to only 50% of their total relief). This allows for likelihood that a certain (but unknown) fraction of the observed valley relief likely pre-dated (and was exploited by) glaciation. Aside from glacial fjord, trough, and valley incision and the associated isostatic response,

the first-order topographic configuration of Greenland is unlikely to have been significantly altered since glacial inception (Alley et al., 2019). Although there remains a significant unresolved debate surrounding the extent of Neogene tectonic uplift in eastern Greenland (Japsen et al., 2013; Pedersen et al., 2012), any such uplift likely predated (and may have been a key precursor of) ice-sheet nucleation (Japsen et al., 2014; Solgaard et al., 2013).



### 2.2.3 Climate forcing

Climatologically, ice-sheet extent and dynamics are highly sensitive to temperature and precipitation (e.g., Dolan et al., 2015; Golledge et al., 2008). Accurately reconstructing and modelling past temperature and precipitation patterns in Greenland is therefore a key community challenge (Haywood, Dowsett, Dolan, et al., 2016). Given the high degree of uncertainty surrounding Greenland's palaeoclimate beyond the most recent glacial-interglacial cycles, we do not attempt a complex coupled general circulation atmosphere / ice-sheet model approach. Instead, we adopted a deliberately simplistic ice-sheet

modelling strategy, whereby we performed experiments to identify broad-scale temperature and precipitation conditions that are (in)capable of producing ice masses with extent and dynamics consistent with the mapped valley geomorphology along the southern and eastern highlands. We then assessed the implications of our findings for the chronology and timing of valley incision.

In our core experiments, we prescribed a range of simple temperature distributions defined by two parameters: the mean annual

air temperature (MAAT) at sea level at 60°N (i.e., the southern tip of Greenland), and the latitudinal gradient (i.e., the rate at which temperature decreases moving northwards; Fig. 3a). For sea level MAAT at 60°N, we used 1°C increments from 2 to 6 °C, representing a range of values higher than the modern-day average for this latitude (~-1°C) (Noël et al., 2019). The modern latitudinal temperature gradient over Greenland is ~0.8 °C/°N (Noël et al., 2019); reduced polar amplification prior to the growth of a continental-scale ice sheet would most likely result in a weaker latitudinal gradient (Burls et al., 2021). We

therefore used a range of gradients from 0.2 to 0.5 °C/°N in 0.1 °C/°N increments. We also assumed that temperature varies with altitude according to a constant moist atmospheric lapse rate of -6.5 °C km$^{-1}$ (Kerr & Sugden, 1994) and tested the sensitivity of the model output to this choice using a stronger lapse rate of -8 °C km$^{-1}$. The temperature field and surface mass balance were updated after every time step to account for changes in ice surface elevation (i.e., the ice-elevation feedback). On top of the MAAT values, we imposed a periodic annual cycle with an amplitude of ±10 °C, comparable to modern-day and

inferred Pliocene annual variability (Ballantyne et al., 2010; Noël et al., 2019).

For precipitation, we used absolute values derived from the multi-model mean of the Pliocene Model Intercomparison Project phase 2 (PlioMIP2) general circulation model output (Haywood et al., 2020). The multi-model mean precipitation is diminished over eastern Greenland due to the presence of a prescribed ice cap in the PRISM4 (Pliocene Research Interpretation and Synoptic Mapping) boundary conditions used in PlioMIP2 (Dowsett et al., 2016). To avoid circularity, this effect was

removed by identifying the amplitude of the air temperature anomaly associated with the prescribed east Greenland ice cap and using a scaling of 5.5 % precipitation change per °C to adjust the multi-model mean precipitation (Nicola et al., 2023). The adjusted precipitation field is characterised by high rates in coastal areas of southeast Greenland (up to 2000 mm yr$^{-1}$), decreasing to low rates (<400 mm yr$^{-1}$) in northern areas (Fig. 3b). For simplicity we assumed constant precipitation throughout the year. We did not use an ensemble of precipitation fields as for MAAT, but instead tested the sensitivity of our results to

precipitation by performing additional experiments with annual precipitation values one standard deviation below and above




the multi-model mean. Temperature and precipitation were used to calculate the surface mass balance (difference between accumulation and ablation). Accumulation is equal to precipitation when air temperatures are below 0°C and decreases linearly to zero at temperatures above 2°C. Ablation was computed using a positive degree-day (PDD) model, which integrates the time where temperatures are above 0°C. We used the PISM default PDD factors, which are based on EISMINT-Greenland

values (Huybrechts, 1998).

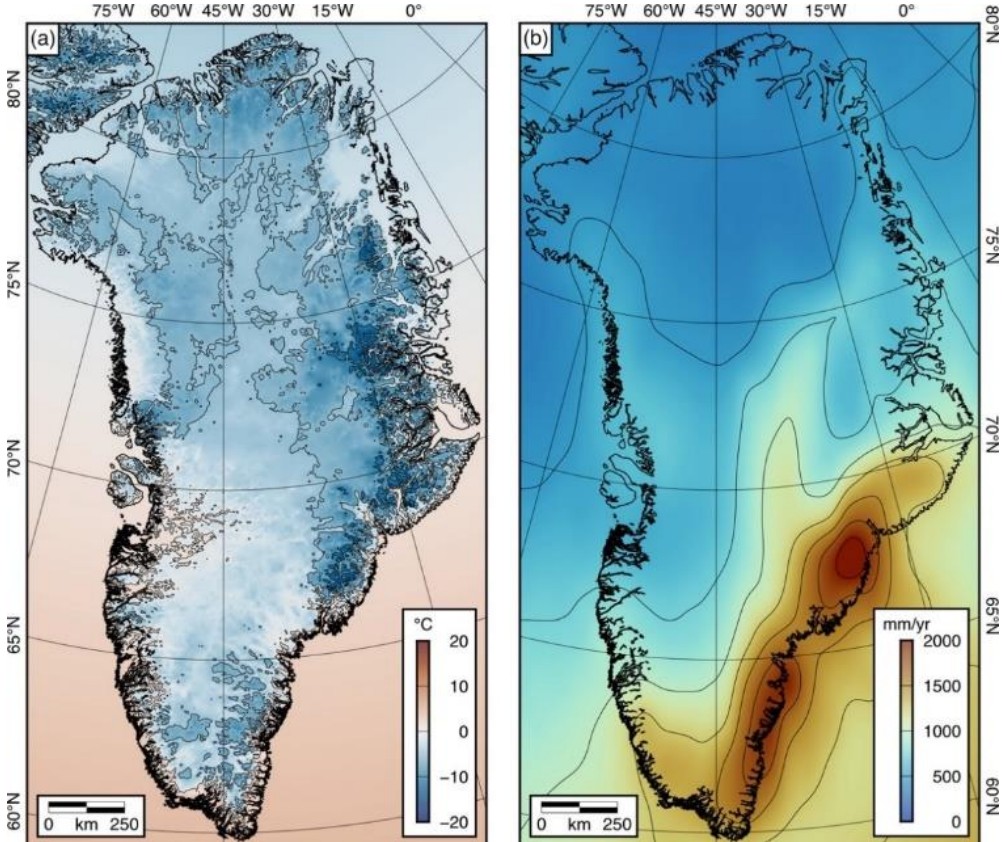

**Figure 3. Baseline temperature and precipitation for ice-sheet model experiments.** (a) Example of a mean annual air temperature (MAAT) field. The MAAT field shown is one of an ensemble of 20 and was constructed assuming a MAAT of +4°C at 60°N, a latitudinal gradient of -0.3 °C/°N, and a vertical lapse rate of -6.5 °C/km. Contour interval is 5°C. (b) Annual precipitation, derived from the PlioMIP2

multi-model mean and adjusted for the removal of the influence of ice on Greenland. Contour interval is 250 mm/yr.

## 3 Results

### 3.1 Subglacial valley networks

Using MODIS MoG imagery, we mapped over 300 subglacial valley segments along the length of the ice-covered highlands of southern Greenland (60–65°N) and eastern Greenland (66–78°N) (Fig. 4a). The valleys can be divided into two first-order



categories: (i) those with a predominantly eastward orientation, which terminate at the modern-day ice margin or enter the fjord systems exposed along the North Atlantic coast and (ii) those with a predominantly westward (inland-facing) orientation (Fig. 4). We traced a central divide separating these two groups of valleys that trends north-south along the spine of the highlands (Fig. 4).

MODIS MoG imagery also reveals that the valleys exhibit complex planform arrangements and often form interconnected networks with sinuous (Fig. 4b), branching/dendritic (Fig. 4c), and radial (Fig. 4d) structures. Some networks contain multiple distinct tributaries, whereas others exhibit a simpler structure with relatively few discernible tributaries (Fig. 4). We note that the ice surface morphology imagery will likely not capture basal topographic features on a scale smaller than the local ice thickness, so is unlikely to resolve the narrower (sub-kilometre), lower-order tributaries within a valley network. In multiple
locations, valleys are crosscut by present-day ice divides and convergent tributaries join in a direction that is oblique to, or opposes, the modern ice flow direction (Fig. 4e).

The presence of these inferred subglacial valleys is confirmed by RES data (Fig. 4g), which show that the textural lineaments observed in the MODIS MoG imagery often correlate with subglacial valleys. Previous comparison of valley cross-profile
morphology with glacial and fluvial valleys elsewhere in the Northern Hemisphere (Paxman, 2023) indicates that 85% of valleys within the highlands are classified as 'glacial' in morphology (i.e., deep, wide, U-shaped, high curvature) and the remainder as 'fluvial' (i.e., shallow, narrow, V-shaped, low curvature). Notably, however, the majority (~72%) of the 'glacial' valleys are presently situated beneath ice that is likely frozen at the bed (cold-based; Fig. S2b) (MacGregor et al., 2022) and therefore non-erosive. The orientations of the coast-facing valleys tend to be strongly aligned with the flow direction of the
modern ice sheet (Fig. 5a) and the mean surface speed of the ice flowing through these valleys is 250 m/yr (Joughin et al., 2018) (Fig. 5b). By contrast, the inland-terminating valleys have a significantly lower mean ice surface flow speed of 20 m/yr (Joughin et al., 2018) (Fig. 4f, 5b) and are typically not aligned with contemporary ice flow; indeed there is a slight preferential misalignment of ~180° (Fig. 5a), implying that ice commonly flows (slowly) up (rather than down) these valleys.





**Figure 4: Subglacial valleys in the eastern and southern highlands.** (a) Valleys identified in MODIS MoG ice surface morphology imagery. Green lines denote inland-facing valleys; purple lines denote coast-facing valleys; dashed white lines delineate the divides between these valleys. Black lines show modern ice divides; blue triangles mark deep ice core sites. Greenland surface elevation is displayed in greyscale (Howat et al., 2022); offshore bathymetry according to the colour scale (Morlighem et al., 2017). (b–e) Valley networks within the four 400 km × 400 km areas indicated in panel a. (f) Ice surface speed (Joughin et al., 2018) (sampled from Fig. S2a) along profile X–X′ over inland-facing valleys (location shown in panel d). (g) Ice-penetrating radargram along profile X–X′ (OIB flight segment 20170422_01). Arrows denote valleys identified in RES survey data (red = 'glacial' classification; blue = 'fluvial' classification) (Paxman, 2023) that are correlated with lineations in MODIS MoG ice surface morphology imagery. Coloured bars denote the likely basal thermal state of the ice sheet (red = likely thawed; white = uncertain; blue = likely frozen) (MacGregor et al., 2022) (sampled from Fig. S2b).



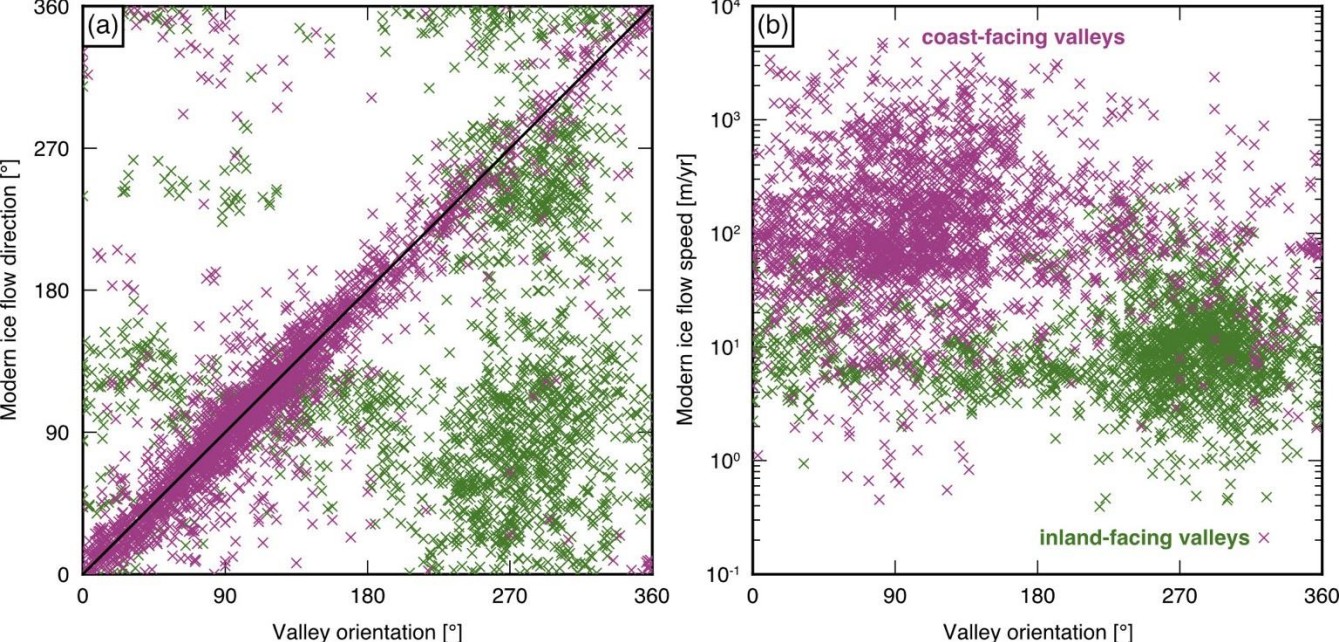

**Figure 5: Relationship between subglacial valleys and modern ice flow.** (a) Valley orientation vs. modern ice surface flow direction (Joughin et al., 2018) for all inland-facing (green) and coast-facing (purple) valleys. Each symbol represents a point within the valley network; the horizontal spacing between adjacent points within all traced valleys is 2 km. Points where both orientations align will plot along the solid line. (b) Valley orientation vs. modern ice surface speed (Joughin et al., 2018).

RES data indicate that 'glacial' valleys are densely clustered in the southern and eastern highlands, whereas 'fluvial' valleys become more common further into the continental interior and in lower-lying gaps between the highlands (Fig. 6a). There is also a small, but non-negligible, fraction of 'fluvial' valleys within the highlands, which varies between 20% and 30% in the southern highlands but is typically below 15% in the eastern highlands (Fig. 6c). Another notable pattern in valley morphology is that the mean depth of valleys in the southern highlands is ~550 m, compared to ~980 m in the eastern highlands (Fig. 6b,d). Very few valleys in the southern highlands exceed 1 km depth, whereas many valleys in the eastern highlands exhibit depths of 1–2 km (Fig. 6b). A dichotomy is also observed between the depths of the valleys on the inland and coastal sides of the highlands, with coast-facing valleys exhibiting a higher mean depth than inland-facing valleys (~1070 m vs. 690 m), as well as a larger interquartile range (Fig. 6e).







**Figure 6: Subglacial valley characteristics in the eastern and southern highlands.** (a) Valleys identified from RES survey data (red = 'glacial'; blue = 'fluvial'). Valleys are overlain on the BedMachine v.5 onshore bed topography, which has been isostatically adjusted for the removal of the modern ice sheet load (Morlighem et al., 2017; Paxman et al., 2022). White lines are divides between coast- and inland-facing valleys; blue triangles mark deep ice core sites. (b) Variations in valley depth (i.e., ridge-to-floor relief) along the southern highlands (SH) and eastern highlands (EH). Geographic extents are shown by boxes in panel a and horizontal arrows in panel b. Background colours denote first-order geological provinces, whose extents are shown in Fig. S1b; KOB = Ketilidian Orogenic Belt; NAC = North Atlantic Craton; RC = Rae Craton; EGVP = East Greenland Volcanic Province; COB = Caledonian Orogenic Belt (Henriksen et al., 2009). (c) Fraction of valleys classified as 'fluvial' within 1° latitudinal bins. (d) Boxplot of the depths of 'glacial' valleys in the SH and EH. (e) Boxplot of the depths of inland- (green) and coast-facing (purple) 'glacial' valleys within both the SH and EH.

The northern and southern limits of the valleys within the southern and eastern highlands are well-defined, with a distinct gap between 65 and 66 °N. The western (inland) limit of the glacial valleys is also relatively well-defined, both in ice surface morphology imagery and RES data (Fig. 4, 6); the coastward limit is more difficult to judge given that the valley networks





typically feed into the exposed fjords along the North Atlantic coast. In the MODIS MoG imagery, the inland-facing valleys often transition from clearly distinguishable to absent over a relatively short horizontal distance. Some surface features that

are likely diagnostic of subglacial valleys are visible further west, but these are much more linear in planform and have previously been interpreted to reflect the upper reaches of palaeo-fluvial drainage pathways that extend from the inland foothills of the eastern highlands to the west coast of the island (Cooper et al., 2016; Cooper et al., 2019a; Paxman et al., 2021). Similarly, RES data show a strong clustering of 'glacial' valleys within the highlands, with a well-defined western margin, beyond which valleys are rarer and those that are observed are more commonly classified as 'fluvial' (Paxman, 2023). Given

the sufficient coverage of RES flight lines (Fig. S1a) and MODIS MoG imagery is not featureless due to proximity to ice divides, we are confident that the observed valley limit is not an artefact of data coverage. For the purposes of comparison with our ice-sheet model results, we constructed polygons demarcating the areas covered by the mapped mountain valley networks, assuming that these valleys can be used to infer the approximate long-term average glacial limit at the time(s) of valley incision (Fig. 7). This exercise was most challenging along the eastern margin of the eastern highlands, where the limit

of the mountain glacial valley networks is ambiguous due to the superimposition of the coastal fjord systems. For simplicity, we traced the eastern limit along the heads of the fjord systems, close to the modern ice margin (Fig. 7).

## 3.2 Simulations of mountain ice growth

Our ice-sheet modelling experiments show that the temperature field that yields an ice configuration that best matches the

glacial valley limit in the eastern highlands is defined by a sea level MAAT of +4 °C at 60°N and a latitudinal MAAT gradient of -0.3 °C/°N (Fig. 7m). These parameters yield a simulated ice sheet that covers a latitudinal range in strong agreement with the mapped valley limits and an inland margin positioned close to the inferred limit (Fig. 7m). The strength of the agreement was quantified by converting both the simulated ice extent and mapped glacial limit into binary fields (where 1 denotes the presence of ice and 0 the absence of ice). We computed the mean absolute deviation between the two binary fields as a measure

of the spatial misfit (i.e., a lower mean absolute deviation signifies a greater number of grid cells in which the model and geomorphological observations agree regarding whether ice is present or absent). The mean absolute deviation confirms this combination of temperature parameters as producing the best-fitting modelled ice sheet in eastern Greenland (Fig. 7u), and also reveals that there is a trade-off between the sea level MAAT at 60°N and the latitudinal MAAT gradient, such that similarly good fits to the mapped glacial valley limit can be achieved via a combined increase in SL MAAT at 60°N and a strengthening

of the latitudinal MAAT gradient, or vice versa (Fig. 7u).

However, the simulated glacial extent under the temperature scenario that yields the best match with the mapped valley limit in the eastern highlands (Fig. 7m) is more restricted in the southern highlands than is implied by the valley limit. An ice mass that best matches the glacial valley limit in southern Greenland is obtained by decreasing the sea level MAAT at 60°N to +3°C and steepening the latitudinal MAAT gradient to -0.4 °C/°N (Fig. 7g). As for the eastern highlands, there is a trade-off between

the two temperature parameters (Fig. 7v). Although high topography is also present in northern Greenland, ice here is absent





or highly restricted in most of our simulations owing to the low precipitation rates in this region (Fig. 3b, 7). Only under the lowest tested temperatures and steepest latitudinal gradients does the development of a large-scale ice sheet occur in northern Greenland (e.g., Fig. 7a).

Sensitivity testing indicates that increasing/decreasing the annual precipitation by one standard deviation from the multi-model mean increases/decreases the size of the modelled ice sheet by an amount approximately equivalent to that achieved by decreasing/increasing the MAAT by 1°C (Fig. S3a-c). Increasing the atmospheric lapse rate to -8 °C km$^{-1}$ predictably causes an increase in simulated highland ice extent, again by an amount comparable to that achieved by decreasing the MAAT by 1°C (Fig. S3d). By contrast, bed topography and geothermal heat flux exert a relatively minor influence on the modelled ice-sheet geometry (Fig. S3e,f). The higher of the two bed topographies intuitively results in a slight increase in modelled ice volume due to the dependence of air temperature upon elevation (Fig. S3e). These sensitivity testing results, combined with the observed trade-off between sea level MAAT at 60°N and the latitudinal MAAT gradient (Fig. 7), suggest that reasonable uncertainty bounds are ±1 °C for the best-fitting sea level MAAT at 60°N and ±0.1 °C/°N for the latitudinal MAAT gradient.







**Figure 7: Ice-sheet model ensemble results.** Panels a–t show simulated steady-state ice thicknesses for a range of combinations of sea level
MAAT at 60°N (columns) and the latitudinal MAAT gradient (rows). In each experiment, ice was grown from scratch on an ice-free
palaeotopography (displayed in greyscale). Solid black outlines show the approximate palaeo-ice extent implied by the mapped valleys (Fig.
4, 6). The sea level equivalent (SLE) of each modelled ice sheet is labelled (modern GrIS SLE = 7.42 m). Panels u and v show the mean
absolute deviation (i.e., misfit) matrices between each simulated ice extent and mapped glacial valley limit for the eastern (u) and southern
(v) highlands (a lower misfit signifies greater domain-wide agreement between the model and geomorphological observations regarding
whether ice is present and absent). The parameter combination that yields an ice configuration most consistent with the mapped glacial limit
in the eastern highlands is a 60°N SL MAAT of +4°C and a latitudinal temperature gradient of -0.3°C / °N (panel m). For the southern
highlands, a lower 60°N SL MAAT of +3°C and a steeper latitudinal gradient of -0.4°C / °N yields the smallest misfit (panel g).





Irrespective of the uncertainties associated with the assumed temperature and precipitation fields, our results demonstrate that ice masses with extents consistent with those implied by the mapped glacial valley networks can be simulated using a relatively

simple numerical ice-sheet model set-up. A single latitudinal temperature gradient yields simulated ice masses that closely match the glacial valley limit along much of the eastern highlands, and a second, cooler, temperature field yields a similarly good fit in the southern highlands. The ice masses that best fit the glacial valley limit in the eastern highlands contains a total volume of $1.7 \times 10^5$ km$^3$ and a sea level equivalent of 0.41 m (Fig. 7m), which is ~6% of that of the modern GrIS (7.42 m) (Morlighem et al., 2017). Given the trade-off between the two temperature parameters (Fig. 7u), a similarly good fit can be

achieved for ice masses with sea level equivalents in the range 0.34 to 0.53 m (Fig. 7i,q). The modelled ice masses that yield the best-fit in the southern highlands contain a larger volume of $7.3 \times 10^5$ km$^3$ and a sea level equivalent 1.82 m, which is ~25% of that of the modern GrIS (with a range of 1.60–2.46 m; Fig. 7c,g,k,v). The implication is that at the time of valley incision in the eastern and southern highlands, barystatic sea level was significantly higher than today, with a contribution from Greenland of between 5 and 7 metres.


We stress that these values represent total Greenland ice volume under the best-fitting climate forcing scenarios. The scenario shown in Fig. 7g contains a greater ice volume than Fig. 7m not because the best-fitting southern highlands ice mass is larger than that of the eastern highlands, but because under the climate that best fits the glacial valley limit in the southern highlands there is a substantial ice cap present in eastern Greenland, and smaller bodies also present in northern Greenland. Indeed, the

SLE of just the best-fitting ice mass in southern Greenland is 0.11 m. The combined SLE of the two best-fitting highland ice masses in isolation is therefore 0.52 m (~7% of that of the modern GrIS). We also note that because of the assumed palaeotopographic configuration, where we adjusted the topography for the flexural response to glacial valley and fjord incision but did not fill in the valleys themselves (section 2.2.2), these modelled ice volumes are likely to be upper estimates, given that there is more accommodation space available within the valleys than may have been the case at the time of mountain

ice growth. Conversely however, given the limits in RES data coverage and model resolution, small-scale (<10 km) basal roughness (which would increase the amount of accommodation space) will be poorly represented in our models.

For the best-fitting ice configurations (Fig. 7g,m), we performed additional high-resolution (5 km) simulations to examine the dynamics of the modelled ice masses in greater detail. These simulations produced ice masses of a very similar configuration

and thickness to (and with total volumes within ±5% of) the lower-resolution (10 km) simulations (Fig. 8a,e), confirming that our first-order findings are not strongly dependent on model resolution. These high-resolution simulations show that ice flow is steered through the subglacial valleys, with flow directions consistent with the mapped valley networks and elevated ice surface velocities through many of the coast- and inland-facing valleys (Fig. 8b,f). Ice divides are established over subglacial ridges (Fig. 8b,f), with major drainage divides running north-south as is implied by the mapped valley network geometry (Fig.

4). Similarly, warm-based ice (which we defined as having a basal melt rate exceeding $10^{-4}$ m/yr) is primarily located within valleys, whereas cold-based (frozen) ice predominates within the interior of the ice masses that are situated over higher terrain



(Fig. 8c,g). Although a quantitative relationship between glacial dynamics and erosion rates remains difficult to establish, it is widely held that erosion predominantly occurs beneath warm-based ice and scales with the basal sliding velocity (Cook et al., 2020; Herman et al., 2015; Koppes et al., 2015). Thus, it is notable that basal sliding velocities are elevated within valleys but
negligible over high terrain, indicating that the erosive potential of these ice masses would have been strongly selective (Fig. 8d,h). These ice masses would therefore have been capable of preserving high topography (peaks and ridges) while contemporaneously incising the intervening valleys. We also note that modelled basal sliding velocity magnitudes are comparable in both the southern and eastern highlands, with the highest sliding velocities typically found on the coast-facing side of the highlands (Fig. 8d,h). The distributions of basal thermal state and sliding velocity are also more diffuse on the
inland side of the highlands compared to the coastal side; this in part likely reflects the fact that many of the inland-facing valleys are not well resolved in the BedMachine digital elevation model due to poorer RES data coverage and the unsuitability of mass conservation techniques in (modern-day) slower-flowing areas, meaning simulated ice flow is not as strongly topographically focussed here as on the coastal side of the highlands.

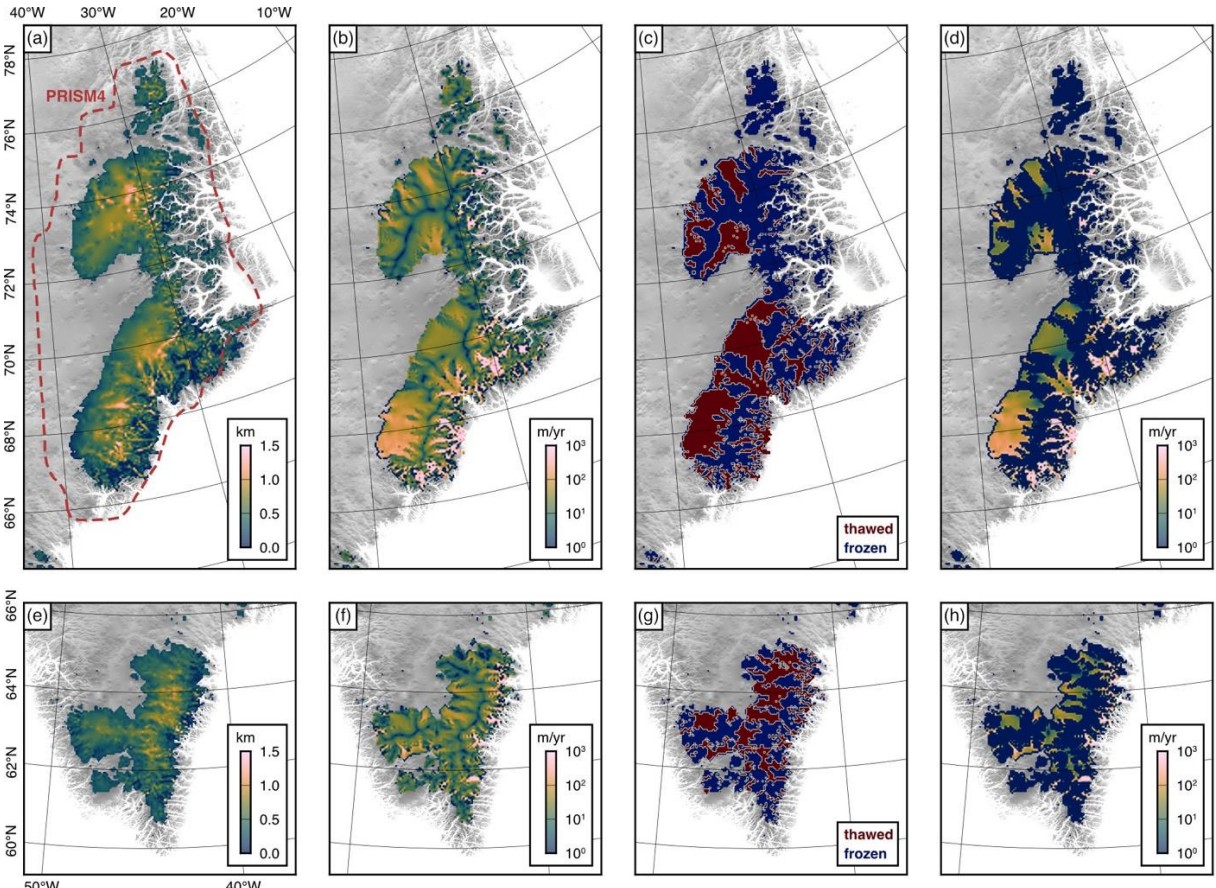

**Figure 8: Dynamics of the modelled ice masses in eastern and southern Greenland.** The simulations shown were performed at 5 km resolution with an otherwise identical model set up to the best-fitting ice configurations shown in Fig. 7m (eastern highlands; top row) and



Fig. 7g (southern highlands; bottom row). (a, e) Ice thickness. (b, f) Ice surface velocity. (c, g) Basal thermal state (thawed = warm-based, frozen = cold-based). (d, h) Basal sliding velocity, which is a proxy for erosive potential. For comparison, the red dashed line in panel a
marks the PRISM4 GrIS extent used in PlioMIP2 (note that PRISM4 assumed an absence of ice in the southern highlands) (Dowsett et al., 2016; Haywood, Dowsett, Dolan, et al., 2016).

## 4 Discussion

### 4.1 Origin of valley networks in the eastern and southern highlands

Despite predominantly resembling glacial valleys in their cross-profile morphology (Paxman, 2023), the valley networks we
have mapped in the highlands of eastern and southern Greenland show several characteristics that are inconsistent with formation beneath the modern continental-scale ice sheet. Many valleys are situated beneath modern slow-moving, cold-based ice, particularly the inland-facing valleys on the western side of the highlands (Fig. 4f,g, 5b). Additionally, valley planforms commonly exhibit intricate structures including sinuous, dendritic, and radial drainage patterns (Fig. 4) that are indicative of convergent/divergent flow through valleys with orientations that are often inconsistent with the modern ice surface velocity
field (Fig. 5a), which overrides the landscape. The observation that numerous valleys are crosscut by modern ice divides (Fig. 4) also indicates that these valleys were not incised by the modern GrIS or by subglacial meltwater at its base. These valley patterns are more consistent with formation by fluvial and/or local-scale glacial erosion. There is a non-negligible fraction of 'fluvial' valleys within the highlands (Fig. 6), indicating that elements of a fluvial landscape have been preserved since glacial inception, most likely facilitated by the long-term presence of cold-based, slow-flowing ice, as is present in the centre of our
simulated ice masses (Fig. 8). We therefore surmise that the mapped valleys most likely represent inherited pre-glacial fluvial networks that were exploited and erosively modified by topographically steered valley glaciers, as part of a smaller ice field earlier in Greenland's glacial history. Modern-day analogues for the inferred palaeoenvironment include locations characterised by topographically steered valley glaciers emanating from central ice fields, such as south Patagonia and British Columbia, Yukon, and Alaska.

The mapped extent and geometry of the palaeo-glacial valleys may therefore reflect the approximate long-term average configuration and dynamics of the former ice mass(es) responsible for their incision (Porter, 1989). However, rather than reflecting a former limit of erosion, an alternative hypothesis is that the distribution of the palaeo-glacial valleys is governed by the modern and/or long-term distribution of non-erosive (i.e., cold-based) ice, and instead reflects a limit of long-term preservation. Comparison of the observed valley network limits with the estimated likely basal thermal state of the modern
GrIS (MacGregor et al., 2022) indicates that, other than at the northern end of the eastern highlands and in the gap between the eastern and southern highlands, the mapped valley limit does not coincide with an abrupt transition from cold- to warm-based ice (Fig. S2b), as would be expected if this limit reflected the extent of landscape preservation by current and former comparable continental-scale ice sheets. Indeed, cold-based, slow-flowing ice extends hundreds of kilometres inland of the limit of the palaeo-glacial network along much of the eastern highlands (MacGregor et al., 2022) (Fig. S2), suggesting it is





unlikely that the observed valley network is the remainder of what was once a much wider signal that was partially removed by subsequent ice-sheet-scale erosion. This conclusion is also supported by the observation that the landscape further into the Greenlandic interior often appears to resemble those of fluvial systems (Bamber et al., 2013; Cooper et al., 2016; Paxman, 2023; Paxman et al., 2021), hinting at long-term landscape preservation.

    A notable observation is that valleys in the eastern highlands are on average almost twice as deeply incised as those in the

southern highlands (Fig. 6b,d). This implies that southern Greenland has experienced a lesser amount of glacial incision integrated over its history than eastern Greenland. Two potential hypotheses to account for this observation are: (i) a lithologically-controlled difference in bedrock erodibility and (ii) a differing glacial history, for example a shorter overall duration of erosive mountain-scale glaciation or a lower average mass turnover (i.e., slower glacier velocities and lower incision rates) in southern Greenland than eastern Greenland.

Geologically, southern Greenland comprises (moving from south to north) the Ketilidian Orogenic Belt (Palaeoproterozoic metasediments and granitoid intrusions deformed during the Palaeoproterozoic Ketilidian Orogeny) and the North Atlantic Craton (Archaean granitic gneiss) (Fig. S1b) (Henriksen et al., 2009). The highlands of eastern Greenland span (from south to north), the Rae Craton (Palaeoproterozoic with reworked Archaean granitic gneiss), the East Greenland Volcanic Province (Palaeogene basalt and mafic intrusions), and the Caledonian Orogenic Belt (Mesoproterozoic metasediments deformed during

the Silurian Caledonian Orogeny) (Fig. S1b) (Henriksen et al., 2009). Notably, the North Atlantic Craton (south Greenland) and Rae Craton (east Greenland) are geologically similar, and would therefore be expected to be characterised by a similar hardness / erodibility (Campforts et al., 2020), but show a strong contrast in valley incision depth (Fig. 6b). Moreover, despite the significant geological variation along the eastern highlands, average and maximum valley depths remain comparable along the full length of the mountain chain, and consistently deeper than those in the southern highlands (Fig. 6b).

These observations indicate that lithology is not a significant control on the spatial variations in valley depth, which may instead be attributable to a contrasting glacial history between the southern and eastern highlands. Our ice-sheet model ensemble results show that in all simulations which exhibit ice growth in the southern highlands there is always ice present in the eastern highlands, but the same is not true in reverse (Fig. 7). Glaciation of the southern highlands requires a cooler island-wide temperature field than the eastern highlands despite the higher modelled precipitation rates (Fig. 3b), owing to the higher

temperatures at lower latitudes (Fig. 3a) and the lower maximum altitude of the southern highlands compared to the eastern highlands (Morlighem et al., 2017). Given that the southern highlands are less likely to be glaciated than the eastern highlands (Fig. 7), it follows that they have likely experienced a shorter duration of glacial incision integrated over Greenland's history, accounting for the lower average valley depths. This would also account for the greater incidence of preserved palaeo-fluvial valleys in the southern highlands (Fig. 6c).



An alternative scenario, that the eastern and southern highlands were occupied by erosive mountain ice for similar durations of time but the ice in eastern Greenland was associated with a higher mass turnover (and therefore greater basal sliding velocities and erosion rates), can likely be discounted given that the elevated palaeo-precipitation rates in southern Greenland indicated by general circulation models (Fig. 3b) would likely result in higher rates of mass accumulation and turnover. Moreover, our simulations do not show a clear difference in modelled ice surface or basal sliding velocities between the

southern and eastern highlands (Fig. 8). Therefore, while ice growth and mountain glacial valley incision may have occurred near-simultaneously along the full length of the eastern highlands (66–78°N) under a single latitudinal temperature gradient (Fig. 8a), we suggest that this pre-dated glaciation in southern Greenland and/or that mountain glaciation in southern Greenland was shorter-lived / more intermittent due to a warmer climate and lower average topography.

The observation that coast-facing valleys are deeper on average than inland-facing valleys (Fig. 6e) is also indicative of a

contrasting incision history between these two groups of valleys. Having discounted a major role for geological variability along the highlands, this second dichotomy may reflect (a) contrasting inherited (pre-glacial) valley depths, and/or (b) a contrasting glacial incision history. Given the origin of the eastern and southern highlands via passive margin uplift following North Atlantic breakup (Bonow et al., 2014; Japsen et al., 2014), fluvial valleys on either side of the central drainage divide would have responded to continental separation and uplift by incising down to base level. Given their proximity to the coast,

antecedent rivers on the east side of the highlands would be expected to have cut deep valleys through the uplifting margin to depths close to sea level, whereas antecedent rivers on the western side of the highlands would have been located several hundred kilometres up-catchment of their own base level (i.e., the west coast) and thus would not have been driven to incise such deep valleys (Beaumont et al., 2000; Cockburn et al., 2000). Although these valleys will have been subsequently excavated and overdeepened by valley glaciers, an inherited disparity in their depths either side of the drainage divide may

have survived to the present day.

This scenario does not preclude the possibility that the coast-facing valleys have also experienced a greater amount of incision beneath fast-flowing, warm-based ice integrated over Greenland's glacial history. For example, both coast- and inland-facing valleys may have been incised by similar amounts during early mountain glaciation, and subsequent continental-scale ice sheets may have contemporaneously preserved the inland-facing valleys beneath cold-based ice (Bierman et al., 2014;

MacGregor et al., 2022), while continuing to deepen the coast-facing valleys that were ideally oriented for exploitation by the warm-based, fast-flowing outlet glaciers that drain outwards from the modern ice-sheet interior (Fig. 5) (Joughin et al., 2018). Faster basal sliding velocities on the coastal side of the highlands during mountain-scale glaciation (Fig. 8), which likely resulted from an increased maritime influence on the climate on the eastern side of the highlands (i.e., elevated precipitation and mass turnover rates; Fig. 3b) could have also contributed to a disparity in palaeo-incision rates (Cook et al., 2020; Seguinot

& Delaney, 2021). A caveat attached to this explanation is that our models may underestimate the basal sliding velocities (and therefore erosion potential) of the inland-facing valleys because many of these features are not well resolved in BedMachine. Some or all of these mechanisms may have worked in concert to leave the coastal valleys generally deeper than those inland.





## 4.2 Timing of landscape modification

While we have ascertained that the landscape of the eastern and southern highlands was significantly modified during one or

more earlier phases of restricted, mountain-scale glaciation, the age of these event(s) cannot be easily determined from the onshore geomorphology alone. In this section, we explore whether our geomorphological mapping and ice-sheet modelling results can be combined with the findings of previous studies to constrain the timing and chronology of mountain valley glaciation in eastern and southern Greenland.

A challenge in addressing this question is that ice sheets and their underlying landscapes are continually evolving; subglacial

topography does not reflect a single 'definitive past configuration' of the ice sheet but is instead better thought of as a palimpsest produced during several glacial stages (Kleman, 1994). Such a landscape may therefore broadly relate to average glacial conditions over multiple glacial cycles (Porter, 1989), although this does not necessarily mean that the topography can be used to directly infer the historical mean or modal glacial conditions (Spagnolo et al., 2022). However, the presence of well-developed and morphologically consistent palaeo-glacial valley networks along the southern and eastern highlands of

Greenland does suggest that ice likely occupied these regions in a configuration conducive to the incision of these valley networks for a protracted timespan. This configuration may not have existed for a single continuous episode, but instead during repeated intervals within a longer window of time, given the typical responsiveness of mountain glaciers to (e.g., orbital) climate variability. Here we focus on a series of time intervals pertinent to Greenland's glacial history and assess the relative likelihood of the valleys having experienced significant and repeated phases of incision during these times.

### 540 4.2.1 The Last Interglacial

The most recent time interval when the GrIS was potentially significantly diminished in extent relative to today is the Last Interglacial (Eemian; MIS 5e; ca. 125 ka), when global mean surface temperatures were 1.5–2°C higher than during the current interglacial period (Clark & Huybers, 2009). Although ice loss at this time was likely non-negligible, numerical ice-sheet models consistently predict that the Eemian GrIS still covered much of central Greenland (Fig. 9a), and global sea level records

do not require an extensive loss of ice from Greenland at this time (Dutton et al., 2015). The identification of Eemian ice in northern Greenland via glacial radiostratigraphy (MacGregor et al., 2015) and in the NEEM, GRIP/GISP2, and DYE-3 deep ice cores (Dahl-Jensen et al., 2013; Suwa et al., 2006; Yau et al., 2016) supports this view, and argues in favour of the persistence of a significantly more extensive ice sheet at this time than the restricted ice masses that are likely to have incised the palaeo-glacial valleys in eastern and southern Greenland (Fig. 8).

### 550 4.2.2 Pleistocene 'super-interglacials'

Cosmogenic nuclide analysis from sediment and bedrock obtained via sub-ice drilling at GISP2 and Camp Century (Fig. 1b) indicates that at least one episode of major ice loss occurred in Greenland at or since ca. 1.1 Ma (Christ et al., 2021; Schaefer et al., 2016), perhaps corresponding to one or more particularly warm and/or long-lasting interglacial periods such as MIS 11



(ca. 430–400 ka) or 31 (ca. 1.09–1.06 Ma). Ice retreat simulations indicate that deglaciation of the Camp Century site
(northwest Greenland; Fig. 1b) during MIS11 would require loss of at least ~20% of the GrIS (Christ et al., 2023), while an
ice-free GISP2 site would necessitate reduction of the GrIS to <10% of its current volume, restricted to isolated ice caps in the
eastern and/or southern highlands (Fyke et al., 2014; Schaefer et al., 2016). These ice core sites are both ice-free in our best-
fitting ice-sheet simulations (Fig. 9). Although the exact chronology remains uncertain, it is therefore plausible that during the
most prominent Pleistocene interglacials (e.g., MIS 11 and 31) the GrIS could have been restricted to a configuration capable
of the incision of the mapped palaeo-glacial valley networks (Fig. 4, 8).

However, an important question is whether the ice sheet existed in this restricted state for a sufficiently long period(s) of time
to account for observed valley depths of 1–2 km. Even allowing for an upper estimate for the combined duration of the MIS
11 and 31 interglacial peaks of ~50 kyr (de Wet et al., 2016) and an assumption that half of the observed valley relief pre-dated
glaciation, this would necessitate average erosion rates of ~10–20 mm/yr to account for the incision of the valleys during these
intervals alone. While estimating long-term glacial erosion rates from ice-sheet model output remains highly uncertain, using
modelled basal sliding velocities of ~100 m/yr (Fig. 8d,h) in a variety of empirical glacial erosion 'laws' (Cook et al., 2020;
Herman et al., 2015; Koppes et al., 2015) gives expected erosion rates on the order of ~2 mm/yr within the valleys. The latter
value is also more consistent with the typical range associated with alpine/polar glaciers (Koppes & Montgomery, 2009; Patton
et al., 2022). Moreover, MIS 11 simulations suggest that while the GrIS was much-diminished, basal ice remained below the
pressure melting point across much of the remaining ice cap (Robinson et al., 2017), helping to preserve the landscape at
GISP2 beneath cold-based, non-erosive ice (Bierman et al., 2014).

The restricted ice caps present during Pleistocene 'super-interglacials' may therefore have contributed to the incision of the
valley networks in eastern and southern Greenland but were likely insufficiently long-lived or erosive to fully account for the
observed relief of these features. Moreover, isostatic calculations indicate that glacial valley and fjord incision in the near-
coastal regions of eastern Greenland must have primarily occurred prior ca. 2.5 Ma (Pedersen et al., 2019). This is because
late Pliocene-early Pleistocene shallow marine sediments are exposed along the shoreline north of Scoresby Sund at elevations
much lower than would be expected if they had been deposited prior to fjord incision and consequently flexurally uplifted by
hundreds of metres (Pedersen et al., 2019). This indicates that fjord incision and concomitant isostatic adjustment must have
largely pre-dated sediment deposition, pointing towards significant glacial incision having commenced in eastern Greenland
prior to the Quaternary.



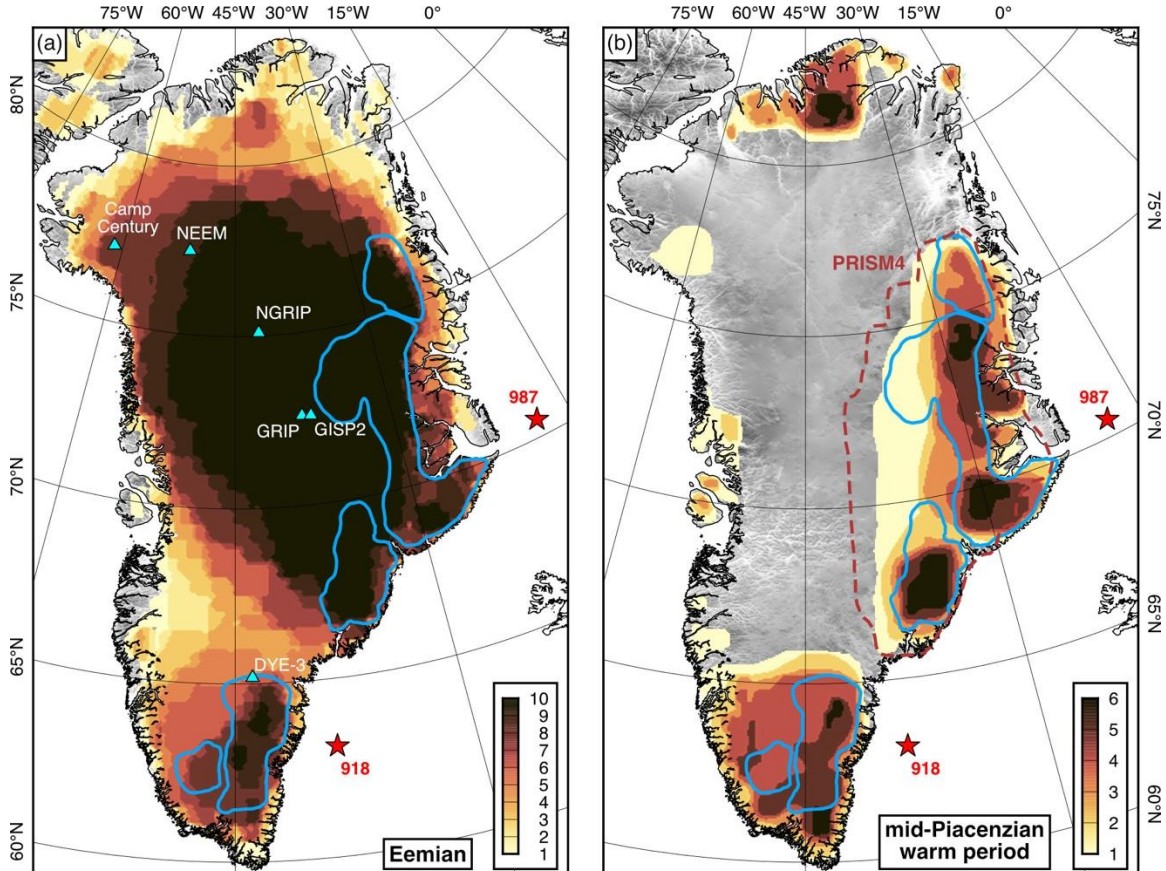

**Figure 9: Heat maps representing simulated GrIS extent during past warmer intervals.** (a) Simulated GrIS minimum extent during the Last Interglacial (Eemian; ca. 125 ka), based on a recent compilation of 10 published ice-sheet model outputs (Plach et al., 2018). These simulations examine the retreat of the GrIS during the Last Interglacial. Colours indicate the number of model simulations that predict the presence of ice at a given location. Light blue triangles mark deep ice cores; red stars mark ODP core sites; blue polygons denote the highland palaeo-glacial valley limits mapped in this study. (b) Simulated GrIS extent during the mid-Piacenzian warm period (ca. 3.26–3.02 Ma), based on the results of 6 different ice-sheet models (Koenig et al., 2015). These simulations examine the growth of the GrIS on an ice-free Greenland using a late Pliocene climatology from the HadAM3 general circulation model. Red dashed line in marks the PRISM4 GrIS extent used in PlioMIP2 (Dowsett et al., 2016; Haywood, Dowsett, Dolan, et al., 2016).

### 4.2.3 Neogene ice growth

If mountain glacial valley incision did not primarily occur during Quaternary interglacial ice-sheet retreat episodes, the implication is that these valley networks primarily reflect early ice-sheet development during the Neogene. A number of modelling studies indicate that in the late Pliocene (ca. 3.6–2.6 Ma) the GrIS nucleated on, and expanded from, the mountains of eastern and southern Greenland (Berends et al., 2019; Contoux et al., 2015; Koenig et al., 2015; Lunt et al., 2009; Solgaard et al., 2011; Tan et al., 2018), although these results do not preclude the presence of upland ice masses prior to the late Pliocene (Bierman et al., 2016). Across multiple ice-sheet models that adopt a mPWP climate forcing from an atmospheric general circulation model in which no GrIS exists *a priori*, the simulated ice sheet is confined to the eastern and southern highlands




(Koenig et al., 2015), with an extent that is broadly consistent with the mapped palaeo-glacial valley limit and our simple ice-sheet modelling results (Fig. 9b).

Data-based constraints on Greenland's climate during the late Pliocene are rare and derive from a small number of sites. On Ellesmere Island (78°N), multiproxy analysis including tree-ring width and oxygen isotopes, palaeo-vegetation composition, and tetraether lipids have been used to infer MAATs of -1°C (± 4°C) during the late Pliocene (Ballantyne et al., 2010; Csank et al., 2011). Similarly, earlier analysis of floral and faunal fossil assemblages from the Kap København Formation in northern Greenland (82°N) provided an indicative estimate for local late Pliocene MAATs of -4°C (Funder et al., 2001). In our ice-

sheet modelling experiments, we obtained an ice sheet with a geometry in closest agreement with the observed glacial valley network extent for a sea level MAAT of +4°C at 60°N and a latitudinal gradient of -0.3 °C/°N (Fig. 7). This implies a sea level MAAT of -2°C at 80°N, which is broadly consistent with these data-based constraints for late Pliocene temperatures in northern Greenland. The best-fitting latitudinal gradient is also broadly consistent with estimated global palaeolatitudinal temperature gradients for the late Miocene and Pliocene, which, while highly uncertain, are typically less than half as steep between 60°N

and 90°N as at the present-day (~-0.8°C/°N) (Bradshaw et al., 2012; Burls et al., 2021; Zhang et al., 2019). The temperature conditions necessary to develop mountain ice masses capable of incising the mapped palaeo-glacial valleys therefore appear to be broadly consistent with those that characterised the late Miocene and/or Pliocene epochs. The inferred global sea level highstand of up to +25 m above modern global mean sea level during the mPWP (Dumitru et al., 2019; Grant et al., 2019) is also consistent with a significantly diminished GrIS at this time (Dutton et al., 2015). The ice masses that best match the extent

of the palaeo-glacial valleys contain ~0.4 m SLE in the eastern highlands and ~0.1 m SLE in the southern highlands (Fig. 7), implying that at the time these valleys were incised, the GrIS contribution to barystatic sea level was as much as ~7 metres.

Offshore records of early Greenlandic glacial development prior to continental-scale ice-sheet formation indicate that glacial material began to appear in North Atlantic sediments during the late Miocene. At ODP Site 918 (southeast Greenland; Fig. 9), episodes of coarse sand, till, diamicton, and periodic ice-rafted debris (IRD) deposition commenced at ca. 7 Ma (Larsen et al.,

1994; St. John & Krissek, 2002). Despite the southerly location of Site 918, sediments here were likely advected southwards by the East Greenland Current, which has been circulating anti-clockwise along the continental shelf since the late Miocene, and were likely sourced from rocks in the highlands of eastern Greenland rather than southern Greenland (Blake-Mizen et al., 2019). Similarly, at ODP Site 987 (east Greenland; Fig. 9), dropstones suggest that ice rafting commenced at ca. 7.5 Ma (Butt et al., 2001). Offshore seismic stratigraphy close to Site 987 also reveals that pre-late Miocene Atlantic sediments lack evidence

of progradation or coarse-grained material, implying an absence of ice at the coast (Pérez et al., 2018). The first progradational glacial unit, indicative of outlet glaciers reaching the coast, is dated at ca. 7 Ma, followed by a broader major glacial advance in the early Pliocene (ca. 5 Ma) (Nielsen & Kuijpers, 2013; Pérez et al., 2018). Glaciers then appear to have retreated during the mPWP, followed by a second major advance at the Pliocene-Pleistocene transition (Pérez et al., 2018).

In our best-fitting ice-sheet model scenario for the eastern highlands (Fig. 7m), the ice sheet reaches the east coast, most

frequently along the southern half of the eastern highlands (Fig. 8a). The extent of the marine-terminating margins of the



modelled ice sheets with extents more restricted than the best-fitting scenario appears to be reduced, although this is difficult to assess robustly given the uncertainties in the palaeotopography. This suggests that, while relatively restricted compared to the modern GrIS, these simulated ice masses were likely sufficiently extensive to reach the coast, at least in certain areas, and could have been responsible for the delivery of glaciogenic sedimentary material and IRD to the continental shelf during the

late Miocene and Pliocene via tidewater glaciers. We note however that mountain glacial valley incision will have commenced prior to the establishment of a marine-terminating margin (Bierman et al., 2016).

The offshore seismic stratigraphy and drill core records therefore indicate that two plausible intervals for mountain glacial valley incision are (i) the late Miocene prior to the large-scale early Pliocene (ca. 5 Ma) margin advance, and/or (ii) subsequent warm periods of the late Pliocene (e.g., the mPWP). It is difficult to determine from the evidence presented here whether

glacial incision primarily occurred during both or only one of these two possible windows. The simplest scenario is that the mapped palaeo-glacial landscape primarily reflects the presence of restricted mountain-scale ice masses during the late Pliocene immediately prior to the onset of continental-scale glaciation (Funder et al., 2001) and consequent establishment of widespread cold-based, non-erosive ice. However, if a marine-terminating ice margin existed (at least intermittently) during the late Miocene / early Pliocene, mountain glacial valley incision likely commenced prior to the late Pliocene. Therefore, on

the available evidence, we simply suggest that glacial valley incision in the eastern and southern highlands primarily occurred early in Greenland's glacial history, most likely in the late Miocene and/or subsequent warm intervals of the late Pliocene. To further resolve which of these two candidate intervals was characterised by the most incision, we recommend future analysis of offshore seismic data and/or acquisition of sediment cores on the east Greenland continental shelf to compare the provenance of Miocene and Pliocene sediments and the timing of pulses of erosion from the eastern and southern highlands.

Following the development of the alpine-style glacial landscape in southern and eastern Greenland over an extended interval (on the order of 100s of kyr to a few Myr), we suggest that the GrIS likely expanded to continental scale comparatively quickly. This inference is made on the basis of (i) ice-sheet modelling that suggests rapid GrIS expansion could have occurred via a surface elevation-mass balance feedback once a key temperature threshold was reached (DeConto et al., 2008), which is also borne out by our own ice-sheet model results (Fig. 7 shows that for every 1°C of cooling, the modelled ice volume

approximately doubles), and (ii) the subglacial geomorphology, which suggests that large parts of the interior landscape have not been strongly modified by glacial erosion (Bamber et al., 2013; Bierman et al., 2014; Cooper et al., 2016), implying that the ice margin (which would be expected to generally be characterised by the fastest ice flow and thus the highest erosion rates) was not situated in these locations (cf. Figs 7c and 7k) for an extended period of time. Much of the landscape in the Greenlandic interior, including the inland-facing valleys in the eastern and southern highlands, would thus have been preserved

beneath a cold-based core of the expanded continental-scale ice sheet. We therefore suggest that the inland-facing side of the eastern highlands in particular may be a promising region in which to find ancient ice in Greenland, on the basis that (a) the topography is generally high, with deep valleys last incised during early mountain glaciation, (b) the ice is cold-based and



slow-moving (Joughin et al., 2018; MacGregor et al., 2022), and (c) ice has likely been persistent for millions of years (Bierman et al., 2016), with potentially the longest continual glacial history in Greenland.

Although the absolute timing remains to be fully resolved, the combined results of our geomorphological mapping and ice-sheet modelling provide an important constraint on the geometry of the GrIS during warm intervals of the late Miocene and/or late Pliocene, and possibly during the strongest Pleistocene interglacial periods. We suggest that the past ice configurations consistent with the incision of the mapped palaeo-glacial valley network (Fig. 8), which contain ~0.4 m SLE in the eastern highlands and ~0.1 m SLE in the southern highlands, could be used to test the outcomes of climate and ice-sheet model
intercomparison projects such as PlioMIP (Haywood, Dowsett, Dolan, et al., 2016). They could also act as an improved, data-constrained, boundary condition for subsequent iterations of global palaeogeography boundary conditions such as PRISM (Dowsett et al., 2016); we highlight that the best-fitting ice configuration in the eastern highlands is significantly less extensive than the ice cap assumed in PRISM4 (Fig. 8a). Moreover, these constraints could be retro-actively applied to existing model intercomparison projects to rule in/out certain simulated past ice sheet scenarios. We also highlight the need for a better
understanding of the chronology of landscape development. This could be facilitated by improved mapping and quantification of offshore sediment volumes and chronologies (e.g., Pedersen et al., 2018), as well as geochemical analysis of the provenance of eroded material entering the North Atlantic (e.g., Blake-Mizen et al., 2019; Cook et al., 2013), which would help better quantify the times at which the ice margin reached the coast or retreated inland along the eastern margin of Greenland.

## 5 Conclusions

In this study we have used a combination of geomorphological mapping and ice-sheet modelling to constrain the behaviour of the Greenland Ice Sheet during past warmer climates. We draw the following conclusions:

1. The highlands of eastern and southern Greenland are incised by a complex network of subglacial valleys that retain signatures of pre-glacial fluvial incision in their observed planform geometry and cross-sectional profile morphology.
The distribution of the subglacial valleys is inconsistent with incision beneath the modern continental-scale GrIS.

2. Instead, the inherited palaeo-fluvial valleys likely steered, and were modified by, the flow of (warm-based, erosive) mountain valley glaciers that drained central high-elevation (cold-based, non-erosive) ice fields during past warmer climates. Our ice-sheet modelling indicates that the mapped palaeo-glacial valley limit in the eastern highlands is consistent with a palaeoclimate characterised by simulated late Pliocene precipitation rates and a simple air
temperature field (defined by a sea level MAAT at 60°N of +4 ± 1 °C, a latitudinal gradient of -0.3 ± 0.1 °C/°N, and a vertical lapse rate of -6.5 °C km$^{-1}$) that is broadly consistent with terrestrial records for the late Pliocene.

3. A further cooling of 1 °C and steepening of the latitudinal gradient by -0.1 °C/°N is required to match the mapped palaeo-glacial valley limit in the southern highlands. Geomorphological mapping also indicates that the landscape of the southern highlands has been less extensively modified by mountain-scale glaciation. We suggest that glaciation





in the eastern highlands likely pre-dated the southern highlands, reflecting the cooler temperatures and higher elevations of this region. The total volume of the simulated ice masses that best match the mapped palaeo-glacial valley limit is ~0.4 m SLE in the eastern highlands and ~0.1 m SLE in the southern highlands, indicating that at the time of valley incision, the GrIS was significantly more restricted in extent than today, with a contribution to barystatic sea level of up to ~7 metres.

4. We suggest that the late Miocene (ca. 7–5 Ma) and late Pliocene (ca. 3.6–2.6 Ma) are the most likely time intervals for the incision of the mapped palaeo-glacial valley network, although we cannot exclude the possibility that the valleys were also incised during particularly warm and/or long-lived 'super-interglacial' intervals of the Pleistocene.

    5. Subsequent transition(s) between a mountain- and continental-scale GrIS did not substantially erode the landscape of the Greenlandic interior, indicating that ice margin advance/retreat occurred relatively quickly and/or non-erosively.

As a result, the inland-facing side of the eastern highlands may contain some of the oldest ice preserved in Greenland.

**Data/Code availability**

The BedMachine version 5 (https://doi.org/10.5067/GMEVBWFLWA7X) and MEaSUREs MODIS MoG version 2 (https://doi.org/10.5067/9ZO79PHOTYE5) datasets were accessed via the National Snow and Ice Data Center. The isostatic

response to the unloading of the modern Greenland Ice Sheet (Paxman et al., 2022) is available via the Arctic Data Center (https://doi.org/10.18739/A2280509Z). The dataset of Greenland subglacial valleys (Paxman, 2023) is available via https://doi.org/10.5281/zenodo.7794565. Data produced in this study, including ESRI shapefiles of the mountain valley network and inferred palaeo-glacial limit and NetCDF files of selected ice-sheet model output, is available via https://doi.org/10.5281/zenodo.10013441.


**Acknowledgements**

GJGP was funded by a Leverhulme Trust Early Career Fellowship, award number ECF-2021-549. MJB was supported by funding received from the European Research Council (ERC) under the European Union's Horizon 2020 research and innovation programme, grant agreement 885205. The authors would like to thank Lauren Burton, Tamara Fletcher, Alan

Haywood, Daniel Hill, Erin McClymont, and Julia Tindall for helpful discussions on late Pliocene Greenland, along with the broader PlioMIP climate modelling community for their ongoing efforts. We also extend thanks to Andreas Plach and Andreas Born for providing the modelled MIS 5e Greenland Ice Sheet extents used to construct the heat map in Fig. 9a. GJGP would like to thank the attendees of the 'Greenland Ice Sheet: lessons from the past' workshop in Bergen, Norway (April 2023), especially Jason Briner, Petra Langebroek, Vivi Pedersen, and Julien Seguinot, for fruitful discussions that helped shape this

manuscript. Figures were prepared using the Generic Mapping Tools (GMT) version 6 software package (Wessel et al., 2019) and colour palettes from Scientific Colour Maps version 7 (Crameri et al., 2020).



**Author contributions**

GJGP conceived the project, performed the analysis, constructed the figures, and drafted the manuscript. SSRJ and MJB provided input on the study scope, methods, and interpretation. AMD provided general circulation model output for the ice-sheet modelling and input on the interpretation of the model results. All authors contributed comments and revisions to the final manuscript.

**Competing interests**

The authors declare that they have no conflict of interest.

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
