# Peer review of "Subglacial valleys preserved in the highlands of south and east Greenland record restricted ice extent during past warmer climates"

_EGUsphere, 2023_

## Author Comment (AC1)

We thank the editor and reviewers for their constructive comments on this manuscript. Below, we explain the changes we have made to the manuscript and address the comments made by Referee #1. Comments made by the reviewer are in *black italics*, and our responses are in red.

RC1 (Anonymous Referee #1)

*L119: The authors state that MODIS MoG imagery records the intensity of the reflection of a satellite-emitted radar signal. This is untrue. MODIS is a passive sensor which records visible/near-visible solar radiation reflections.*

We thank the reviewer for catching this error. These lines have been amended to the following:

*"MODIS is a passive sensor that records the intensity of (near-)visible solar radiation reflections; the MoG surface morphology map is derived by high-pass filtering of red-light MODIS images (Haran et al., 2018). MoG surface morphology imagery therefore provides a semi-quantitative approximation of the reflectivity of the ice surface, which depends on the slope / curvature."*

*L125: 'Conversely' is confusing here. Both the preceding sentence and this one focus on shortcomings of the MoG approach, whereas conversely would suggest that we're about to be told something about what it is good for. 'Furthermore' instead?*

We agree and have changed 'conversely' to 'furthermore' as suggested.

*L136: Please add a further reference that cites the OIB project.*

References to both MacGregor et al. (2021; Reviews of Geophysics) and Paden et al., (2019; NSIDC) have been added here.

*L139-140: Either expand this methods explanation or remove incompletely - there isn't enough information to judge whether the fact that an ML approach was used is important to the present study or not. If it is, expand why, otherwise I suggest removing this methodological detail.*

The reviewer is correct that this methodology was not carried out in the present study, but instead refers to the previous study of Paxman (2023). We have therefore removed the excess detail and the text is now simplified as follows:

*"This dataset (Paxman, 2023) also contains quantitative metrics of valley cross-profile morphology, including depth, width, V-shapedness, and curvature, as well as classifications of valleys as either 'glacial' or 'fluvial' based on their morphological similarity to glacial or fluvial valleys observed elsewhere in the Northern Hemisphere. For the purposes of this study, we only examined the morphometrics of valley profiles whose classification is associated with high confidence (for more information the reader is referred to Paxman, 2023)."*

*L277: add a reference to Fig. 4g for concerning the thermal state analysis.*

A reference to Fig. 4g has been added here.

*L279: (subjective) - I suggest starting a new paragraph here for clarity.*

Agreed; a new paragraph has been started here.

*L332: is 'mapped mountain valley networks' missing the term 'glacial'? (to distinguish from fluvial)*

It is, thank you for catching this. We have inserted the word 'glacial' here.

*L480: The first part of this sentence is really the conclusion of the previous paragraph, so perhaps would be better off added there. Then the paragraphs will match points (a) and (b) introduced in L466-469.*

This is a good suggestion and we have followed the suggestion of the reviewer and shifted this sentence to the end of the previous paragraph.

*L494: would 'also' result in higher rates of mass accumulation and turnover 'there', or similar (i.e. is the intended meaning that the conditions which enable higher rates of mass accumulation in turnover in the EH also cause the ~same conditions in the SH?)*

This sentence was intended to point out that the high precipitation rates in the SH would likely translate into higher rates of turnover than in the EH (all other things being equal), which argues against the possible scenario that greater valley depths in the EH were caused by higher mass turnover rates / flow velocities there than in the SH. We have rephrased the section to make this clearer:

*"An alternative scenario is that the eastern and southern highlands were occupied by erosive mountain ice for similar durations of time but the ice in eastern Greenland was associated*

*with higher rates of mass turnover (and therefore greater basal sliding velocities and erosion rates). However, this possibility can likely be discounted because the elevated palaeo-precipitation rates in southern Greenland that are consistently indicated by general circulation models (Fig. 3b) would likely result in higher rates of mass accumulation and turnover here than in eastern Greenland (all other things being equal)."*

L572-580: I struggled to understand this paragraph. This might be my shortcomings in being able to 'imagine' isostasy, but I think nonetheless that some rephrasing would be beneficial.

We appreciate that this text was not clear to readers less familiar with isostasy. We have therefore rephrased this entire paragraph to make the logic easier to follow:

*"Moreover, isostatic calculations indicate that glacial valley and fjord incision in the near-coastal regions of eastern Greenland must have primarily occurred prior to ca. 2.5 Ma (Pedersen et al., 2019). This is because incision of these deep valley and fjord systems would be expected to have driven hundreds of metres of flexural uplift of the adjacent coastal areas via erosional unloading, but late Pliocene-early Pleistocene (ca. 2.5 Ma) shallow marine sediments are exposed along the shoreline and have not experienced significant uplift (Pedersen et al., 2019. This indicates that fjord incision and concomitant isostatic adjustment must have largely pre-dated the deposition of these sediments, pointing towards selective glacial valley and fjord incision having largely occurred prior to the Quaternary in eastern Greenland."*

---

## Author Comment (AC2)

We thank the editor and reviewers for their constructive comments on this manuscript. Below, we explain the changes we have made to the manuscript and address the comments made by Referee #2. Comments made by the reviewer are in *black italics*, and our responses are in red.

RC2 (Henry Patton)

*L194: This process for producing a pre-Quaternary topography is unclear to me - why are the refilled subglacial valleys included in the isostatic correction if they are subsequently left open? Are any eroded sediments from troughs/marine sectors on the adjacent shelf included in the isostatic response?*

This is an important question, and we appreciate that we had not explained this clearly. The choice of topography to use in our ice-sheet model experiments was a pragmatic one. We wanted to ensure the highlands were adjusted to pre-glacial elevations so we can get the elevation-climate controls correct in the model, which meant correcting for the isostatic effect of coastal fjord and inland valley incision. However, when it comes to simulating ice flow, if the valleys/fjords were filled with rock, we would not be able to simulate the realistic, topographically steered, flow shown in Fig. 8. To clarify these points, we have separated this paragraph and reworded it as follows:

*"For the erosional unloading correction, we followed the approach described by Medvedev et al. (2013) and Pedersen et al. (2019), whereby the thickness of glacially eroded material was estimated by subtracting the ice-free topography from an accordant surface interpolated between the plateaux and peaks that separate the fjords and valleys. Erosion of the glacial troughs on the adjacent continental shelf was also accounted for. We computed the isostatic response to this incision using an elastic plate model with a laterally variable effective elastic thickness (Paxman et al., 2021; Steffen et al., 2018), and subtracted this correction from the rebounded BedMachine v.5 digital elevation model. This means that the mountain peaks and plateaux (which are assumed to have not been eroded since widespread glaciation) were lowered to their estimated pre-glacial elevations, which was necessary for examining elevation-climate controls in our simulations of early mountain ice growth. The magnitude of the flexural response to fjord and valley incision is 300–600 m along the highlands (Paxman et al., 2021). However, as soon as glaciation commences in our model experiments, it is necessary for the valleys themselves to be left open (i.e., instantaneously incised) in the digital elevation model, enabling us to simulate realistic spatial patterns of ice flow. Indeed, it is reasonable to assume that the modern-day valleys would have at least been partially incised by rivers prior to glaciation. While this means the bed topography boundary condition used in our models will not have existed at a single point in time, it acts as a representative state for*

*the period of early mountain glaciation and allows a trade-off between ensuring that the elevation-climate controls on ice nucleation are correct while also allowing the influence of valleys on ice flow to be simulated."*

To accompany this revised text, we have created an additional figure for the supplement (new Fig. S2; see below), which shows the estimated distribution of erosion and computed flexural response. The only alternatives to the approach we have adopted would be to either: (i) include an erosion law in the ice-sheet model, which would be a complex and poorly constrained task that is unlikely to create a valley network consistent with the one we see today, or (ii) interpolate valley incision through time, but there is currently an absence of firm offshore constraints from around Greenland on the temporal evolution of incision rates. We also emphasise that we performed a sensitivity test with the flexural adjustment reduced by 50% and found that this did not have a significant impact on modelled ice volume (see the original Fig. S3e in the supplement) and we also discuss the (minor) implications of this assumption for total simulated ice volume (original lines 396-401).

[Figure]

Fig. S2. Flexural response to valley, fjord, and trough incision. (a) Estimated eroded thickness due to incision. (b) Computed flexural uplift resulting from erosional unloading. This field is subtracted from the ice-free bed topography to provide the reconstructed bed elevation used in our ice-sheet modelling experiments.

*L344: I think some further clarification on how you calculate these MAD values would be useful e.g., by showing the equation used. I presume it's something like (Σ | (Actual) – (Forecast)|)n, but are you keeping the domain size constant when comparing the two binary fields? What is the region being compared? For example, the MAD values in 7u for b&c look identical but the model misfit in terms of extent in reality is very different.*

We have updated section 3.2 to include the equation used to calculate the MAD. The domain size is held constant when comparing the two binary fields, and the region being compared is now described in this paragraph and illustrated in the updated Fig. 7 (see below), which now contains a legend to assist with interpretation.

[Figure]

Fig. 7 (updated).

*L443: Rather state the fraction than force the reader to lookup a figure.*

The relevant fraction (15%) has been added here.

*L475: Given that ~30% of the glacial valleys are not cold based today and are seemingly being eroded by the present-day ice configuration (Figure 5) does this depth contrast between east and south still hold if comparing just the current cold-based valleys? From Fig 6e it seems these coast-facing valleys of the eastern highlands add some skew.*

This contrast does indeed still hold if comparing just the valleys beneath non-erosive ice. This has been clarified by inserting the following sentence:

"This contrast is also apparent if only the ~72% of valleys that are not experiencing significant erosion beneath the modern GrIS are considered; the mean depths of this subset of valleys are ~570 m in the southern highlands and ~1040 m in the eastern highlands."

*L478: An interesting observation on consistent maximum valley depths - possibly related to negative feedbacks related to sediment evacuation? (cf. Fig 13 Patton et al., 2016).*

This is an interesting possibility, and we have added a sentence along with the reference to mention it:

"We suggest that the apparent upper limit of ~2 km on valley depths in the eastern highlands (Fig. 6b) may reflect negative feedbacks related to sediment evacuation that act to slow and stabilise the depth of glacial overdeepenings (Patton et al., 2016)."

*L517: This asymmetric long-term development - coast v inland - is well-recognised on the Norwegian glaciated passive margin too, and a similar hypothesis was put forward by Kleman et al., 2008 (section 4.1), and Hall et al. (2013), during mountain-scale glaciation, and could be useful context here.*

We thank the reviewer for suggesting these additional references, which provide an important analogue and add useful context to our discussion. We have inserted a sentence, including both references, to make this point:

"A similar hypothesis has also been proposed to explain the asymmetric incision depths observed on the coast- and inland-facing sides of the conjugate Norwegian glaciated passive margin (Hall et al., 2013; Kleman et al., 2008)."

*L544: Arguably I would not have included the PDD-driven models in this Fig 9a given the conclusions of the Plach 2018 paper, which also strengthens your argument here.*

The 'heat map' in Fig. 9a was constructed using the results of 10 previous ice-sheet modelling studies that are compiled in Plach et al. (2018; their Fig. 2) but were not results from Plach et al.'s own analysis. While Plach et al. (2018) have a clear critique of PDD approaches, we are simply interested here in exploring the broad patterns of previous model output, so have elected not to sub-select from the 10 models shown. We have therefore left our Fig. 9a unchanged but have reworded the caption to explain more clearly what these simulations are and where they have been obtained from:

*"Simulated GrIS minimum extent during the Last Interglacial (Eemian; ca. 125 ka), based on a recent compilation of 10 previously published ice-sheet model outputs compiled in Fig. 2 of Plach et al. (2018)."*

We also note that a similar 'heat map' is shown in Fig. 4b of Haywood et al. (2019; Earth Systems and Environment) and shows a very similar pattern to our Fig. 9a, but the Plach et al. (2018) paper had the advantage of having model output that were openly available.